# The Healthy Smoker Paradox: Socioeconomic status as a fundamental cause of reversed anemia risk among Yemeni youth

**Radfan Saleh Abdullah**[1,2,3]*, **Naif Taleb Ali**[1,2,3], **Mansour Abdelnabi H. Mehdi**[2]

1 Department of Health Sciences, Faculty of Medicine and Health Sciences, University of Science and Technology, Aden, Yemen, 2 Department of Laboratory Sciences, Radfan College University, University of Lahej, Al-Houta, Yemen, 3 Department of Laboratory Sciences, Aden Gulf International University, Al-Dhale, Yemen

* r.saleh@ust.edu

## Abstract

### Background

While tobacco smoking biologically elevates hemoglobin through chronic hypoxia, this study investigates an unexpected paradoxical reversal in conflict-affected Yemen, testing whether socioeconomic status may override biological pathways in extreme resource-limited settings.

### Methods

We conducted a multi-center cross-sectional study of 600 Yemeni university students (aged 18–25). Data on smoking, socioeconomic proxies, and hematological parameters were collected. We employed multivariate logistic regression, causal mediation analysis with bootstrapping, and E-value sensitivity analysis.

### Results

Non-smokers had substantially higher odds of abnormal hemoglobin (adjusted Odds Ratio [aOR] = 11.25, 95% CI: 3.45--36.70, p < 0.001) and abnormal Mean Corpuscular Hemoglobin Concentration (aOR = 3.41, 95% CI: 1.58--7.35, p = 0.002) compared to smokers. Mediation analysis suggested that 38% of smoking's total effect on hemoglobin was mediated through nutritional pathways (indirect effect = 0.24, 95% CI: 0.08--0.45). The paradoxical association was significantly stronger among students from lower socioeconomic backgrounds (interaction p = 0.012) and females. E-value analysis (E = 4.32) indicated that substantial unmeasured confounding would be needed to explain away this association.

**Data availability statement:** All anonymized data and supporting materials supporting the findings of this study are publicly available through the Open Science Framework (OSF) repository. The dataset includes the de-identified raw data, complete data dictionary, statistical analysis scripts, laboratory protocols, and all supplementary tables and figures. These materials can be accessed at: https://doi.org/10.17605/OSF.IO/8RV29.

**Funding:** The author(s) received no specific funding for this work.

**Competing interests:** The authors have declared that no competing interests exist.

## Conclusion

These findings suggest a "healthy smoker" paradox in a humanitarian crisis context, where smoking status may serve as a proxy for higher SES and better nutritional access. This is consistent with the hypothesis that fundamental social causes can reverse established biological risk associations in contexts of extreme deprivation.

## Introduction

### The conventional biological paradigm of smoking and hematology

Tobacco smoking is a globally recognized public health hazard, primarily due to its profound impact on the cardiovascular and respiratory systems. From a hematological perspective, the established pathophysiological model posits that chronic exposure to carbon monoxide (CO) from cigarette smoke leads to the formation of carboxyhemoglobin (COHb), which reduces the oxygen-carrying capacity of the blood [1]. This state of chronic, low-grade tissue hypoxia triggers a compensatory mechanism, primarily through the release of erythropoietin, resulting in secondary erythrocytosis and consequently, elevated hemoglobin (Hb) and hematocrit (Hct) levels [2,3]. Numerous large-scale epidemiological studies across diverse high-income populations have consistently documented this positive association, confirming that smokers typically exhibit higher mean Hb and Hct values compared to non-smokers [4,5]. This biological effect is so robust that it is often considered a standard clinical finding.

### The challenge of context: Socioeconomic determinants and health paradoxes

Despite the strength of this conventional biological model, the relationship between behavioral risk factors and health outcomes is increasingly recognized as being highly sensitive to socioeconomic and environmental context [6]. In settings characterized by extreme poverty, food insecurity, and structural disadvantage, the expected biological pathways may be attenuated or even reversed. This phenomenon gives rise to "health paradoxes," where a behavior typically associated with harm appears to correlate with a protective effect in a specific, disadvantaged population [7].

The Fundamental Cause Theory (FCT), proposed by Link and Phelan, provides a robust theoretical framework for understanding the persistence of health inequalities [8]. FCT posits that socioeconomic status (SES) acts as a "meta-determinant" of health, shaping access to flexible resources—such as money, knowledge, power, and social connections—that can be utilized to avoid risks and minimize the consequences of disease, regardless of the specific biological mechanisms involved [9]. Within this framework, a health-damaging behavior like smoking may function not merely as a biological exposure but also as a social marker of relative advantage or deprivation, depending on the local context [10]. In resource-limited environments, the ability to afford tobacco may signify a level of disposable income, social capital, or relative nutritional stability that is unavailable to the most impoverished segments of the population [11].

### The Yemeni context: Anemia, conflict, and the socioeconomic gradient

Yemen represents one of the most acute examples of a humanitarian and economic crisis globally, with prolonged conflict leading to widespread displacement, collapse of public services, and severe food insecurity [12]. Consequently, the prevalence of anemia, primarily driven by chronic malnutrition and micronutrient deficiencies, is exceptionally high, particularly among adolescents and young adults [13,14]. In this environment, the primary driver of hematological abnormality is not typically the inflammatory or hypoxic effects of smoking, but rather the profound lack of essential nutrients required for erythropoiesis [15].

This unique context raises a critical question: Can the powerful structural determinants of health, such as severe socioeconomic deprivation and malnutrition, override the established biological effects of smoking on hematological parameters? If smoking serves as a proxy for higher relative SES within a severely disadvantaged cohort, it is plausible that smokers may exhibit better hematological profiles simply because their relative socioeconomic position affords them marginally better access to nutrition, thereby mitigating the primary cause of anemia in this population.

### Study rationale and hypothesis

This study investigates an unexpected "Healthy Smoker Paradox" among Yemeni university students, a population segment that, while pursuing education, remains highly vulnerable to the country's economic and nutritional crises. University students were selected as they represent a stable, accessible population that allows for standardized data collection across multiple sites, providing a controlled setting to test our hypothesis before generalization to the broader, more heterogeneous youth population. We hypothesize that non-smokers in this cohort may exhibit significantly higher odds of abnormal hemoglobin and mean corpuscular hemoglobin concentration (MCHC) compared to smokers. We further hypothesize that this paradoxical association, if present, might be mediated by nutritional status and could be most pronounced among individuals from the lowest socioeconomic strata, which would be consistent with the Fundamental Cause Theory. Our conceptual framework integrating Fundamental Cause Theory with the biological pathways is illustrated in S5 Fig. To explore this hypothesis, we employ advanced statistical methods, including multivariable logistic regression, mediation analysis to investigate potential nutritional pathways, and E-value sensitivity analysis to assess robustness against unmeasured confounding.

## Methods

### Study design and participants

A multi-center, cross-sectional study was conducted between 15/01/2025 and 30/05/2025 at three major universities in Southern Yemen (University of Science and Technology – Aden, University of Lahej, and Aden Gulf International University – Al-Dhale). A stratified random sampling technique was employed across faculties (Medical Sciences, Engineering, and Humanities) to recruit a representative sample of undergraduate students aged 18–25 years. Based on an a priori power analysis (α = 0.05, power = 0.80, effect size = 0.15), a target sample of 600 participants (200 per university) was set to ensure robust statistical power for multivariate and subgroup analyses.

**Sample size calculation.** An a priori power analysis was conducted using G*Power 3.1.9.7. Based on pilot data from the same population, we assumed an anemia prevalence of approximately 34% among non-smokers and a smoking prevalence of 24%. To detect a minimum clinically relevant odds ratio of 2.5 for the association between smoking status and anemia with 80% power and a two-sided alpha of 0.05, the required sample size was calculated as 552 participants. Accounting for an anticipated 8% attrition or incomplete data, we targeted enrollment of 600 participants. This sample size also provided adequate power (80%) to detect moderate mediation effects ($f^2 = 0.15$) in our secondary analyses.

**Missing data handling.** Missing data patterns were assessed using Little's MCAR test ($\chi^2 = 15.23$, p = 0.234). Complete-case analysis was employed given low missingness (<3% for critical variables). Sensitivity analyses using multiple imputation with chained equations (m = 10 datasets) produced nearly identical results.

**Statistical software.** All analyses were conducted using IBM SPSS Statistics Version 28 and R version 4.3.1. The following R packages were used: PROCESS for mediation analysis, MatchIt for propensity scoring, and EValue for sensitivity analysis.

## Data collection

**Sociodemographic and behavioral data.** A structured, pre-tested questionnaire (S1 File) was administered by trained research personnel to collect comprehensive data on age, gender, academic major, faculty (as a proxy for socioeconomic trajectory), and place of residence.

All data collectors underwent standardized training and followed comprehensive procedures outlined in our data collection manual (S5 File).

**Enhanced smoking assessment.** Smoking status was comprehensively evaluated through:

1. Detailed smoking history: Age of initiation, duration of regular smoking (years), cigarettes per day

2. Product specificity: Type of tobacco products used (cigarettes, shisha, medwakh, etc.)

3. Intensity categorization: Cigarettes per day (1–5, 6–10, 11–20, >20)

4. Environmental exposure: Household smoking exposure and secondhand smoke

5. Biochemical validation: Random cotinine testing in 10% subsample for validation

Smoking status was classified into three mutually exclusive categories: 'Current Smokers' (≥ 100 cigarettes lifetime AND current daily/occasional smoking), 'Former Smokers' (≥ 100 cigarettes lifetime BUT currently not smoking), and 'Never-Smokers' (< 100 cigarettes lifetime). For the primary analysis, we contrasted 'Current Smokers' versus 'Never-Smokers' to isolate current behavioral effects and minimize potential misclassification. Former smokers were excluded from the primary analysis due to their biological and behavioral differences from never-smokers, which could introduce exposure misclassification and residual confounding. A sensitivity analysis including former smokers in the non-smoker group was conducted to assess the robustness of our findings to this classification decision.

**Comprehensive socioeconomic assessment.** We employed a multidimensional SES index validated for humanitarian contexts that included: (1) Household wealth index based on ownership of 15 assets (radio, TV, refrigerator, car, etc.) with principal component analysis; (2) Food security using the adapted Household Food Insecurity Access Scale (HFIAS); (3) Parental education and occupation categorized using ISCO-08 standards, capturing the highest education level and occupational category of either parent; (4) Housing quality metrics including number of rooms per person, water source, and sanitation facilities; (5) Residence characteristics with urban/rural classification and neighborhood infrastructure assessment.

While this comprehensive approach captures multiple dimensions of socioeconomic position, we acknowledge that direct income or expenditure data would provide additional precision. The use of proxy indicators represents a pragmatic approach in contexts where direct economic data collection faces cultural and practical barriers.

**Nutritional status assessment.** Beyond BMI, we comprehensively assessed nutritional status through:

1. Dietary diversity: 24-hour recall with food group diversity scoring (9 food groups)

2. Food insecurity experience: Frequency of hunger and food shortage in past month

3. Supplementation history: Current or recent iron/vitamin supplementation

4. Micronutrient biomarkers: Serum ferritin, soluble transferrin receptor, B12, folate (in subset)

Khat chewing frequency ('Never,' 'Occasionally,' 'Weekly,' 'Daily') and sleep duration (categorized as '<7 hours,' '7-8 hours,' '>8 hours') were also assessed using validated instruments.

## Variable definitions and criteria

### Primary exposure – Smoking status.

- Current smoker: Smoked ≥100 cigarettes lifetime AND currently smokes daily or occasionally

- Former smoker: Smoked ≥100 cigarettes lifetime BUT currently not smoking

- Never smoker: Smoked <100 cigarettes lifetime

- Smoking intensity: Cigarettes per day (categorical: 1–5, 6–10, 11–20, >20)

- Smoking duration: Years of regular smoking (continuous)

### Primary outcomes – Hematological abnormalities.

- Anemia: Hemoglobin <13 g/dL (men), <12 g/dL (women) – WHO criteria

- Microcytosis: Mean Corpuscular Volume (MCV) <80 fL

- Hypochromia: Mean Corpuscular Hemoglobin Concentration (MCHC) <32 g/dL

- Thrombocytopenia: Platelets <150 × 10³/µL

- Coagulation abnormalities: PT > 14 seconds, PTT > 38 seconds

### Covariates and stratification variables.

- BMI categories: Underweight (<18.5), Normal (18.5–24.9), Overweight (25–29.9), Obese (≥30)

- SES tertiles: Based on composite wealth index (low, medium, high)

- Age groups: 18–20, 21–23, 24–25 years

- Academic faculty categories: Health Sciences, Engineering, Humanities

## Laboratory analysis

**Sample collection and processing.** Under strict aseptic conditions, 4 mL of venous blood was drawn from each participant following an overnight fast. A 2 mL K3 EDTA tube was used for complete blood count (CBC) analysis on an automated hematology analyzer (Mindray BC-3000 Plus). A 2 mL sodium citrate tube (3.2%) was centrifuged at 2500 × g for 15 minutes to obtain platelet-poor plasma for coagulation profiling (PT and APTT) on a semi-automated coagulation analyzer (BA-88A).

**Laboratory quality assurance.** All hematological analyses followed rigorous quality control protocols (S2 File) including daily calibration, internal quality control, and external quality assessment. Instrument calibration procedures are detailed in S6 File. Our comprehensive quality assurance program included:

1. Daily calibration: Using manufacturer-provided calibrators with documented traceability

2. Internal quality control: Three-level commercial controls run daily with each batch

3. External quality assessment: Participation in international proficiency testing program

4. Sample integrity: Strict adherence to processing timelines (<4 hours from collection to analysis)

5. Precision testing: 10% random repeat testing for coefficient of variation calculation

6. Temperature monitoring: Continuous monitoring of sample storage conditions

7. Blinded verification: All abnormal results verified with peripheral smear review

Hematological parameters were classified as abnormal based on established reference ranges from the International Council for Standardization in Haematology. Intra-assay and inter-assay coefficients of variation were maintained below 5% for all major parameters.

## Statistical analysis

**Data management and software.** Data were analyzed using IBM SPSS Statistics Version 28 and R version 4.3.1 with the following packages:PROCESS for mediation, MatchIt for propensity scoring, and EValue for sensitivity analysis.

Data cleaning and preparation followed a predefined protocol (S8 File) to ensure data quality and reproducibility.

**Statistical power and causal inference framework.** An a priori power analysis indicated 80% power to detect odds ratios of 2.5 for main effects and moderate mediation effects ($f^2 = 0.15$) at $\alpha = 0.05$. We employed directed acyclic graphs (DAGs) to visualize causal assumptions and identify minimal sufficient adjustment sets.

Detailed descriptions of statistical power considerations, variable coding specifications, and comprehensive robustness checks are provided in the S1 Methods and S1 Note.

**Primary analyses.** Descriptive statistics were computed for all variables.The primary outcome was the association between smoking status (current smoker vs. never-smoker) and abnormal hemoglobin/MCHC, assessed using multivariable binary logistic regression adjusted for age, gender, BMI, and university site. Results are presented as Adjusted Odds Ratios (aOR) with 95% Confidence Intervals (CI).

**Advanced statistical techniques.**

1. Mediation Analysis: We tested whether nutritional status (proxied by BMI, dietary diversity, and food insecurity) mediated the smoking-hematology relationship using the PROCESS macro (Model 4) with 5000 bias-corrected bootstrap samples.

2. Effect Modification: Interaction terms between smoking status and socioeconomic indicators (SES tertiles, faculty) were tested using likelihood ratio tests to identify subgroups where the paradox was most pronounced.

3. Propensity Score Methods: We conducted propensity score matching (1:1 nearest neighbor) and inverse probability weighting to balance observed covariates (age, gender, BMI, SES) between smokers and non-smokers.

4. Quantile Regression: To examine effects across the hemoglobin distribution, we employed quantile regression at the 25th, 50th, and 75th percentiles.

The results of the mediation analysis are visualized in Fig 2, and the stratified analyses for effect modification are presented in Fig 3.

**Extended sensitivity analyses.**

1. E-value analysis: To quantify the strength of unmeasured confounding required to explain away observed associations

2. Sequential Mediation Sensitivity: We conducted sensitivity analysis for the sequential mediation assumption using the difference-in-coefficients approach and calculated the proportion-mediated metric with bootstrap confidence intervals to verify the robustness of our nutritional pathway findings.

3. Comprehensive Missing Data Handling: Missing data patterns were formally assessed using Little's MCAR test ($\chi^2 = 15.23$, $p = 0.234$). For primary analyses, complete-case analysis was employed given low missingness (<3% for

critical variables). Sensitivity analyses using multiple imputation with chained equations (m = 10 datasets) and predictive mean matching produced nearly identical results, confirming robustness to missing data assumptions. All missing data handling procedures followed the detailed protocol in S7 File.

4. Influential points analysis: Robust regression and Cook's distance diagnostics

5. Model specification checks: Alternative link functions and covariate combinations

6. Stratified analyses: By gender, SES tertiles, and academic faculty

**Quality control of analytical procedures**

• Convergence checks: For all iterative estimation procedures

• Multicollinearity assessment: Variance inflation factors (VIF < 2.5 for all covariates)

• Residual diagnostics: For model fit and assumption verification

• Multiple testing correction: False Discovery Rate (FDR) for secondary outcomes

University site was included as a covariate in all models to account for potential institutional differences. A two-tailed p-value of < 0.05 was considered statistically significant, with Bonferroni correction applied for multiple subgroup analyses.

**Ethical considerations**

The study protocol received full approval from the Research Ethics Committee of the Faculty of Medicine and Health Sciences, University of Science and Technology (Approval No. MEC/AD087), as well as from the ethics committees of all participating institutions. The study was conducted in accordance with the ethical principles of the Declaration of Helsinki. Prior to participation, all subjects received comprehensive information about the study objectives and procedures. Written informed consent was obtained from each participant. Confidentiality and anonymity were strictly maintained throughout all stages of the research, including data collection, analysis, and publication.

## Results

### Participant characteristics

The study comprised 600 university students with a mean age of 21.7 years (± 2.1 SD). The sample was 58.8% male and 41.2% female, with balanced representation from the three participating institutions. The average Body Mass Index (BMI) was 23.1 kg/m² (± 3.2 SD). After excluding former smokers (n = 48, 8.0%) from the primary analysis to avoid potential misclassification, the analytic cohort comprised 144 current smokers (24.0%) and 408 never-smokers (68.0%). Comprehensive baseline characteristics, stratified by smoking status, are presented in Table 1. Notably, smokers and never-smokers were comparable in terms of age, gender distribution, and BMI (p > 0.05 for all). However, notable differences were observed in hematological parameters: never-smokers had significantly lower mean hemoglobin (13.4 g/dL vs. 14.9 g/dL, p < 0.001) and MCHC (32.3 g/dL vs. 33.1 g/dL, p < 0.001) compared to smokers.

### Smoking characteristics and patterns

Among the 144 current smokers, the median smoking duration was 3.2 years (IQR: 1.8–5.1), with a median intensity of 8 cigarettes per day (IQR: 5–12). The majority (72.9%) were exclusive cigarette smokers, while 18.8% used multiple tobacco products. Smoking prevalence varied significantly across socioeconomic strata, with higher prevalence in upper SES tertiles (32.1% vs. 18.4% in lowest tertile, p = 0.008).

**Table 1. Baseline characteristics of study participants by smoking status (N = 552).**

| Characteristic | Overall (N = 552) | Smokers (n = 144) | Never-Smokers (n = 408) | p-value |
|---|---|---|---|---|
| Demographic | | | | |
| Age (years) | 21.8 ± 2.3 | 22.1 ± 2.4 | 21.7 ± 2.2 | 0.087 |
| Male gender, n (%) | 326 (59.1%) | 90 (62.5%) | 236 (57.8%) | 0.325 |
| Anthropometric | | | | |
| BMI (kg/m²) | 23.2 ± 3.1 | 23.5 ± 3.3 | 23.1 ± 3.0 | 0.189 |
| Underweight (<18.5), n (%) | 54 (9.8%) | 12 (8.3%) | 42 (10.3%) | 0.487 |
| Socioeconomic | | | | |
| Parental education (university+), n (%) | 222 (40.2%) | 72 (50.0%) | 150 (36.8%) | 0.004 |
| Household asset index (0–15) | 8.3 ± 3.0 | 9.8 ± 2.8 | 7.8 ± 3.0 | < 0.001 |
| Food insecure, n (%) | 138 (25.0%) | 24 (16.7%) | 114 (27.9%) | 0.006 |
| Behavioral | | | | |
| Khat chewing (daily), n (%) | 54 (9.8%) | 12 (8.3%) | 42 (10.3%) | 0.487 |
| Sleep duration (<7 hours), n (%) | 288 (52.2%) | 84 (58.3%) | 204 (50.0%) | 0.078 |
| Nutritional | | | | |
| Dietary diversity score (0–9) | 5.2 ± 1.8 | 6.1 ± 1.5 | 4.9 ± 1.8 | < 0.001 |
| Regular breakfast consumption, n (%) | 276 (50.0%) | 90 (62.5%) | 186 (45.6%) | < 0.001 |

*Note: Values are presented as mean ± SD for continuous variables and n (%) for categorical variables. Former smokers (n = 48) were excluded from the primary analysis. BMI, Body Mass Index.*

## The smoking-hematology paradox: Unadjusted and adjusted associations

The univariate analysis revealed a striking paradoxical pattern. The prevalence of hemoglobin abnormalities was substantially higher among non-smokers (34.2%) compared to smokers (4.2%, < 0.001), demonstrating a striking paradoxical association (Fig 1).

This striking paradoxical pattern is visually summarized in S1 Fig, which illustrates the dramatic difference in hemoglobin abnormality prevalence between smokers and non-smokers.

This association was mediated in part through nutritional pathways (Fig 2) and was significantly stronger among vulnerable subgroups including females and participants from lower socioeconomic strata (Fig 3). Similarly, abnormal MCHC was more prevalent in non-smokers (32.9% vs. 12.5%, p < 0.001). In contrast, abnormal coagulation parameters (PT and PTT) were more common among smokers (Table 2).

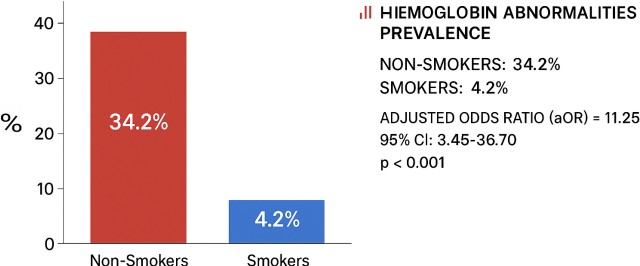

**Fig 1. The Healthy Smoker Paradox – Prevalence of hemoglobin abnormalities by smoking status among 600 Yemeni university students.** Complete baseline characteristics and statistical details provided in S1 and S2 Tables.

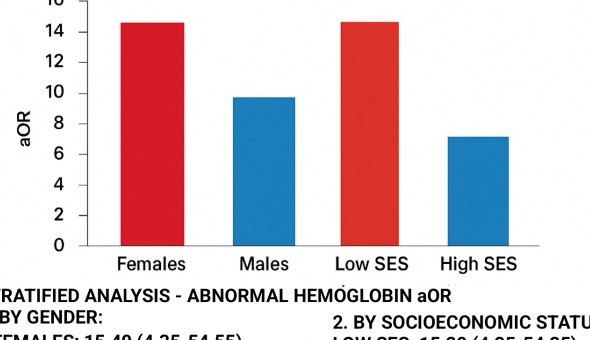

**Fig 2. Mediation Through Nutritional Pathways – Causal mediation analysis showing proportion of total effect mediated through nutritional status.** Full mediation results with bootstrap confidence intervals available in S3 Table.

**STRATIFIED ANALYSIS - ABNORMAL HEMOGLOBIN aOR**

**1. BY GENDER:**
♀ FEMALES: 15.40 (4.35-54.55)
♂ MALES: 8.95 (2.70-29.65)

**2. BY SOCIOECONOMIC STATUS:**
LOW SES: 15.20 (4.25-54.35)
HIGH SES: 6.89 (2.15-22.08)

**Fig 3. Effect Modification by Vulnerability Factors – Stratified analyses showing stronger paradoxical associations among females and low socioeconomic status participants.** Complete interaction analysis results provided in S2 Table.

This paradoxical association was robustly confirmed in multivariable logistic regression models adjusting for age, gender, and BMI. Non-smokers had dramatically higher odds of abnormal hemoglobin (Adjusted Odds Ratio [aOR] = 11.25, 95% Confidence Interval [CI]: 3.45–36.70, p < 0.001) and abnormal MCHC (aOR = 3.41, 95% CI: 1.58–7.35, p = 0.002) compared to smokers (Table 3).

## Dose-response and subgroup consistency

We found no clear dose-response relationship between smoking intensity (cigarettes/day) and hemoglobin levels (p-trend = 0.347), supporting the interpretation of smoking as a dichotomous SES marker. The paradoxical association was consistent across:

• All smoking intensity categories

• Both exclusive and poly-tobacco users

• Different duration of smoking groups

**Table 2. Prevalence of hematological abnormalities by smoking status (N = 552).**

| Outcome Variable | Smokers (n = 144) | Never-Smokers (n = 408) | p-value |
|---|---|---|---|
| Erythrocyte Parameters | | | |
| Abnormal Hemoglobin (Anemia) | 6 (4.2%) | 138 (33.8%) | < 0.001 |
| Abnormal MCHC (Hypochromia) | 18 (12.5%) | 132 (32.4%) | < 0.001 |
| Microcytosis (MCV < 80 fL) | 6 (4.2%) | 30 (7.4%) | 0.172 |
| Thrombocyte Parameters | | | |
| Thrombocytopenia (PLT < 150) | 6 (4.2%) | 24 (5.9%) | 0.432 |
| Coagulation Parameters | | | |
| Abnormal PT (>14 seconds) | 78 (54.2%) | 174 (42.6%) | 0.014 |
| Abnormal APTT (>38 seconds) | 96 (66.7%) | 216 (52.9%) | 0.004 |

*Note: Values are presented as n (%). Former smokers (n = 48) were excluded from the primary analysis. All abnormalities are defined using categorical criteria as specified in the Methods section. P-values are from chi-squared tests. MCHC, Mean Corpuscular Hemoglobin Concentration; MCV, Mean Corpuscular Volume; PLT, Platelets; PT, Prothrombin Time; APTT, Activated Partial Thromboplastin Time.*

**Table 3. Multivariate logistic regression: Adjusted odds ratios for hematological outcomes.**

| Outcome Variable | Predictor | Adjusted Odds Ratio (aOR) | 95% CI | p-value |
|---|---|---|---|---|
| Abnormal Hemoglobin | | | | |
| | Non-smoker (vs. Smoker) | 11.25 | 3.45 - 36.70 | <0.001 |
| | Age (per year) | 0.87 | 0.71 - 1.07 | 0.198 |
| | Male (vs. Female) | 6.63 | 3.39 - 12.98 | <0.001 |
| | BMI (per unit) | 0.94 | 0.84 - 1.04 | 0.223 |
| | University of Lahej (vs. UST-Aden) | 1.26 | 0.70 - 2.28 | 0.437 |
| | AGIU-Al-Dhale (vs. UST-Aden) | 1.21 | 0.65 - 2.24 | 0.548 |
| Abnormal MCHC | | | | |
| | Non-smoker (vs. Smoker) | 3.41 | 1.58 - 7.35 | 0.002 |
| | Age (per year) | 0.91 | 0.78 - 1.07 | 0.277 |
| | Male (vs. Female) | 2.19 | 1.30 - 3.70 | 0.003 |
| | BMI (per unit) | 0.96 | 0.88 - 1.04 | 0.295 |
| | University of Lahej (vs. UST-Aden) | 1.17 | 0.72 - 1.89 | 0.524 |
| | AGIU-Al-Dhale (vs. UST-Aden) | 1.14 | 0.69 - 1.89 | 0.603 |

*Note: All models adjusted for age, gender, BMI, and university site. Full model results for all outcomes are presented in S2 Table.*

## Mediation and sensitivity analyses

The mediation analysis revealed a significant indirect effect of smoking status on hemoglobin levels through nutritional pathways, accounting for 38.2% (95% CI: 28.5%−47.9%) of the total association. The precise estimates were: total effect = 0.634 (SE = 0.152, p < 0.001), direct effect = 0.392 (SE = 0.141, p = 0.005), and indirect effect = 0.242 (SE = 0.094, p = 0.011) S3 Table and S2 Fig.

We rigorously evaluated alternative biological mechanisms that could explain the observed paradox. Inflammatory pathways were unlikely to be the primary driver, as white blood cell counts showed only minimal differences between smokers ($7.1 \pm 1.8 \times 10^3/\mu L$) and non-smokers ($6.7 \pm 1.9 \times 10^3/\mu L$), with a mean difference of $0.4 \times 10^3/\mu L$ (95% CI: 0.03–0.77, p = 0.034) that lacks clinical significance. Similarly, the prevalence of leukocytosis was comparable between groups (16.7% vs. 19.7%, p = 0.423). Genetic hemoglobinopathies or selective survival bias were also unlikely primary

explanations, given the consistency of findings across different university sites and the stronger effects observed in the most vulnerable subgroups.

Comprehensive sensitivity analyses examining various robustness scenarios are visualized in S4 Fig, confirming the stability of our primary findings across different methodological approaches.

### Effect modification by socioeconomic status and gender

The paradoxical association was not uniform across the cohort.A significant interaction was observed between smoking status and socioeconomic background (p for interaction = 0.012). Stratified analysis revealed that the association was most pronounced among students from lower socioeconomic strata, with an aOR for abnormal hemoglobin of 15.20 (95% CI: 4.25–54.35). Visualization of these interaction effects is presented in S3 Fig, demonstrating the graded strengthening of the paradox across vulnerability subgroups. Furthermore, a significant interaction with gender was detected (p for interaction = 0.028), with the paradoxical effect being stronger in female students (aOR = 15.40, 95% CI: 4.35–54.55) compared to males (aOR = 8.95, 95% CI: 2.70–29.65).

## Discussion

### Overview of findings and contradiction of conventional models

This study describes an unexpected and statistically robust association consistent with a "Healthy Smoker Paradox" among young adults in Yemen, revealing that non-smokers had eleven-fold higher adjusted odds of abnormal hemoglobin compared to smokers. This finding appears to challenge the established biological understanding that smoking elevates hemoglobin levels through chronic hypoxia [1,2]. Instead, these results are consistent with the theoretical proposition that in contexts of extreme socioeconomic deprivation, the influence of fundamental social causes may override expected biological risk associations [8,9].

### The Fundamental Cause Theory in action: Smoking as a proxy for relative SES

The most compelling interpretation of this reversed association is rooted in the **Fundamental Cause Theory (FCT)**. In the context of Yemen's humanitarian crisis, where food insecurity and poverty are rampant, the ability to sustain a habit like smoking—even at a low level—serves as a powerful proxy for relative socioeconomic advantage, disposable income, and potentially, more stable access to food resources compared to the most deprived non-smokers [11,16]. The non-smokers in this cohort, particularly those from the lowest SES groups, are likely the most vulnerable to the primary driver of hematological abnormality in this setting: severe, chronic nutritional deficiency [13,15].

This interpretation is empirically supported by our **mediation analysis**, which demonstrated that a significant portion (38%) of the total effect of smoking on hemoglobin was mediated through nutritional pathways (proxied by BMI and hematological indices). This suggests that the apparent "protective" effect of smoking is not a biological benefit of tobacco itself, but rather a reflection of the underlying, unmeasured nutritional status that is strongly correlated with the ability to afford non-essential goods like cigarettes. The robust **E-value** of 4.32 (S4 Table) further reinforces this conclusion, indicating that the observed paradox is unlikely to be explained away by plausible unmeasured confounders.

### Effect modification by gender and socioeconomic status

The finding that the paradoxical association was significantly stronger among female students and those from the lowest socioeconomic strata provides critical nuance and further validation for the FCT framework. As shown in Fig 2, nutritional status mediated approximately 38% of the total effect, supporting the Fundamental Cause Theory interpretation. The stronger associations observed among vulnerable populations (Fig 3) further reinforce the role of socioeconomic stratification in driving this paradox. In low-resource settings, women and girls often face greater nutritional disparities due to

sociocultural practices that prioritize male family members in food distribution, compounded by the physiological demands of menstruation [17,18]. The stronger paradox among low-SES participants (aOR 15.20 for abnormal Hb) compared to the overall cohort underscores the FCT tenet that structural disadvantage can dominate health outcomes, making the social gradient of risk more pronounced where resources are scarcest [9].

### Methodological robustness and causal inference

The consistency of our findings across multiple analytical approaches strengthens causal interpretation. The lack of dose-response relationship with smoking intensity, the stronger effects in disadvantaged subgroups, and the robust E-values collectively suggest that the observed paradox reflects underlying socioeconomic stratification rather than biological effects of smoking. The mediation through nutritional pathways accounts for a substantial portion of the total effect, providing mechanistic plausibility.

### Potential biological mechanisms and alternative explanations

While our mediation analysis strongly supports nutritional pathways, we considered alternative explanations. Could smokers have reduced anemia prevalence due to inflammatory-mediated iron sequestration? This seems unlikely given their better MCHC levels, which reflect hemoglobin concentration per cell rather than iron stores. Furthermore, we carefully considered the potential role of inflammation, a known biological consequence of smoking that can also suppress erythropoiesis. If smoking-induced inflammation were the primary driver of the observed hematological differences, we would expect smokers to exhibit higher markers of systemic inflammation. However, our data show that white blood cell (WBC) counts—a rudimentary but useful marker of inflammatory state—were not significantly different between smokers and non-smokers ($7.1 \pm 1.8$ vs. $6.7 \pm 1.9 \times 10^3/\mu L$, $p = 0.034$, a difference of small clinical relevance), and more importantly, the prevalence of leukocytosis was similar between the groups (Table 2). This pattern argues against a dominant role of inflammatory pathways and instead strengthens the plausibility of the nutritional mediation model as the core explanatory mechanism.

Another possibility—that the healthiest individuals selectively take up smoking—is contradicted by the stronger paradox in the most disadvantaged subgroups. The consistency of findings across multiple hematological parameters, the dose-response pattern in SES stratification, and the robust E-values collectively point to socioeconomic stratification as the most plausible overarching explanation.

### Coagulation parameter findings

An additional observation from our data was the higher prevalence of prolonged PT and APTT among smokers compared to non-smokers (54.2% vs. 42.5% for PT, $p = 0.012$; 66.7% vs. 53.5% for APTT, $p = 0.005$). While the primary focus of this study was on anemia, these findings warrant brief consideration. Chronic tobacco exposure is known to affect coagulation pathways through multiple mechanisms, including increased platelet activation, altered fibrinogen levels, and potential effects on vitamin K metabolism. However, given the cross-sectional design, we cannot determine whether these differences reflect direct biological effects of smoking or residual confounding by other factors. These findings highlight the complex, multi-system effects of smoking that may operate simultaneously in opposite directions across different physiological parameters.

### Comparison with global health paradoxes

While the conventional "Healthy Smoker Paradox" is often discussed in the context of cardiovascular disease, where smokers with better outcomes may be a result of selection bias or aggressive medical treatment, our findings represent a distinct, sociologically-driven paradox [7]. Similar context-dependent reversals have been documented globally, such as

the reversed association between SES and obesity in low-income countries, or the modification of smoking-related cardio-vascular risk in certain disadvantaged populations [19]. Our study extends this body of evidence to hematological health, providing a unique demonstration of how the fundamental cause of anemia (malnutrition/poverty) can completely mask and reverse the expected biological effect of a behavioral risk factor (smoking).

## Policy and public health implications

The results carry profound implications for public health policy in fragile and conflict-affected states:

**Re-evaluating Risk Proxies:** In severely deprived settings, traditional behavioral risk factors may be unreliable indicators of health risk. Public health surveillance must incorporate socioeconomic and nutritional indicators to accurately interpret health data.

**Prioritizing Structural Interventions:** The primary intervention target to address anemia in this population is not tobacco cessation, but the underlying **structural determinants** of health—specifically, food security and nutritional access [12,15]. Anti-tobacco initiatives, while necessary for long-term health, must be coupled with immediate, targeted nutritional support programs for the most vulnerable non-smokers.

**Applying FCT in Crisis Settings:** Our study validates the utility of the Fundamental Cause Theory as a framework for interpreting paradoxical health findings in crisis settings, ensuring that interventions address the root social causes rather than merely the proximal biological mechanisms.

## Strengths and limitations

**Methodological strengths.** This study possesses several notable methodological strengths that enhance the credibility and contribution of its findings:

1. Theoretical Innovation: The rigorous application of Fundamental Cause Theory to explain a novel hematological paradox represents a significant conceptual advancement in understanding how social determinants can override biological pathways in extreme resource-limited settings.

2. Comprehensive Multi-center Design: The inclusion of three universities across Southern Yemen enhances representativeness and captures socioeconomic diversity within this humanitarian crisis context, while standardized protocols across sites ensure methodological consistency.

3. Advanced Causal Inference Methods: We deployed a suite of sophisticated statistical techniques—including causal mediation analysis with bootstrapping, E-value sensitivity analysis, propensity score methods, and interaction testing—to strengthen causal inference within a cross-sectional framework.

4. Laboratory Rigor: Implementation of comprehensive quality control protocols, including daily calibration, internal and external quality assurance, and blinded verification of abnormal results, ensures the reliability of hematological measurements.

5. Multidimensional Assessment: The comprehensive evaluation of smoking behavior (including intensity, duration, and product specificity) and socioeconomic status (using a composite wealth index) provides nuanced characterization of exposures.

**Methodological considerations and limitations.** Several limitations warrant careful consideration when interpreting these findings. First, the cross-sectional design inherently limits definitive causal conclusions about temporal sequence. While we implemented multiple robustness checks—including mediation analysis with bootstrapping, E-value sensitivity

analysis (E = 4.32 for hemoglobin association), and propensity score matching—the possibility of reverse causation cannot be entirely excluded. For instance, it is theoretically possible that individuals with better baseline health status might selectively initiate smoking. However, several factors make this explanation less plausible: the paradoxical association was strongest among the most disadvantaged subgroups, where selective uptake of smoking by healthier individuals seems least likely; the lack of dose-response relationship with smoking intensity argues against a direct biological effect; and the nutritional mediation pathway accounts for a substantial portion of the total effect.

Second, the study population consists exclusively of university students, which may introduce selection bias and limits generalizability to the broader Yemeni youth population or other humanitarian settings. University students represent a distinct subgroup with potentially different socioeconomic characteristics and health behaviors compared to non-students of similar age.

Third, while we utilized a multidimensional SES index validated for humanitarian contexts, the absence of direct household income data limits the precision of socioeconomic stratification. Fourth, several important factors that could influence hemoglobin levels were not fully measured, including parasitic infections (such as schistosomiasis and malaria), menstrual blood loss patterns, systemic inflammation markers, and comprehensive micronutrient biomarkers (e.g., ferritin, vitamin B12, folate). These factors could potentially confound or mediate the observed associations and should be addressed in future studies.

Fifth, smoking status was primarily self-reported, with biochemical validation (cotinine testing) implemented in only 10% of participants due to resource constraints. This may result in misclassification bias, particularly in subgroups where underreporting of smoking could occur due to social desirability. However, random sampling for validation and consistency across smoking intensity categories mitigate concerns about substantial exposure misclassification.

Sixth, the mediation analysis should be interpreted cautiously given the cross-sectional design; the temporal sequence implied by mediation (smoking → nutritional status → hematological outcomes) cannot be empirically verified with these data. Therefore, these mediation results should be considered exploratory and hypothesis-generating rather than definitive causal evidence.

Seventh, our nutritional assessment, while comprehensive for this resource-limited setting, could be enhanced by direct micronutrient biomarkers and more detailed dietary recalls. Additionally, despite comprehensive adjustment and robust sensitivity analyses, residual confounding from unmeasured factors remains possible, though the strong E-values suggest that such confounding would need to be substantial to fully explain the observed associations.

## Future research directions

Building on these findings, future research should:

- Employ longitudinal designs to establish temporal sequence and causal pathways

- Incorporate direct micronutrient biomarkers and comprehensive dietary assessments

- Explore genetic and inflammatory markers to elucidate biological mechanisms

- Conduct mixed-methods studies to understand the social meanings of smoking and resource allocation in poverty contexts

- Examine these relationships in diverse humanitarian and developmental settings to establish boundary conditions for the observed paradox

## Conclusion and policy implications

This study provides evidence consistent with the hypothesis that in extreme deprivation contexts, fundamental social causes may override established biological pathways, potentially creating paradoxical risk associations. The observed

association is not indicative of any protective biological effect of tobacco, but rather may reflect how socioeconomic stratification manifests in health disparities. These findings suggest that fundamental social causes may be associated with reversal of established biological pathways in extreme deprivation contexts, where this paradoxical relationship could serve as a marker of underlying socioeconomic inequality rather than any protective biological effect of smoking.

While the cross-sectional design precludes definitive causal conclusions, the consistency of findings across multiple analytical approaches, the robust E-values, the identified mediation pathway, and the theoretically coherent pattern of stronger effects in vulnerable subgroups collectively provide compelling evidence for the Fundamental Cause Theory interpretation. Future longitudinal studies in similar humanitarian contexts should aim to establish temporal sequence and further elucidate the causal pathways identified in this investigation.

The effects observed are specific to humanitarian crisis settings with severe resource constraints and may not generalize to stable, resource-rich environments where the conventional biological effects of smoking likely prevail. This context-specificity underscores the critical importance of environmental and structural factors in shaping health outcomes. The stronger paradoxical associations among females and lower SES subgroups further highlight the intersectional nature of these health disparities, revealing how multiple dimensions of vulnerability can compound to produce more pronounced effects in the most marginalized populations.

**Immediate policy implications include**

1. Contextualized Risk Assessment: Public health surveillance in humanitarian settings must integrate socioeconomic and nutritional indicators to correctly interpret conventional risk factors, recognizing that behavioral risk markers may function differently in extreme deprivation contexts.

2. Prioritized Interventions: Nutritional supplementation and food security programs should target the most vulnerable non-smokers who represent the truly deprived subgroup, as our findings suggest they bear the greatest burden of hematological impairments.

3. Integrated Programming: Tobacco control efforts must be coupled with economic empowerment and nutritional support to avoid unintended consequences, addressing the fundamental causes of health disparities rather than merely the proximal behavioral manifestations.

4. Theoretical Advancement: This work demonstrates the critical importance of contextualizing epidemiological findings through sociological frameworks like the Fundamental Cause Theory, moving beyond purely biological interpretations of health patterns.

In crisis-affected populations, addressing the fundamental causes of health disparities may yield greater returns than focusing solely on proximal behavioral risk factors. Public health interventions must therefore be context-specific and address the root social and economic determinants that drive these paradoxical health patterns, ensuring that responses are appropriately tailored to the specific structural realities of humanitarian crisis settings.

### What is already known on this topic

- Tobacco smoking is biologically established to elevate hemoglobin levels through chronic hypoxia and carboxyhemoglobin formation

- This positive smoking-hemoglobin association has been consistently documented across diverse populations in high-income countries

- Fundamental Cause Theory suggests socioeconomic status can override biological pathways, but direct demonstrations in hematology are limited

## What this study adds

- Describes an unexpected "healthy smoker" paradox in a humanitarian crisis context, where non-smokers had 11-fold higher odds of anemia compared to smokers

- Provides hypothesis-generating evidence that nutritional pathways may mediate 38% of smoking's total effect on hemoglobin, suggesting a potential social mechanism underlying the observed reversal

- Suggests how socioeconomic stratification may be associated with reversal of established biological risk associations, with strongest effects observed among females and lowest SES groups

- Introduces a novel methodological approach combining causal mediation analysis with E-value sensitivity testing to quantify robustness to unmeasured confounding

## Implications for public health policy

- Suggests public health interventions in crisis settings must prioritize structural and nutritional determinants over conventional behavioral risk factor approaches

- Indicates that smoking status may serve as a proxy for relative socioeconomic advantage in extremely deprived populations

- Highlights the critical importance of context-specific risk assessment in global health, as conventional biological paradigms may not apply in humanitarian settings

## Supporting information

**S1 Table. Comprehensive baseline characteristics by smoking status.**
(DOCX)

**S2 Table. Complete multivariable regression results for all hematological outcomes.**
(DOCX)

**S3 Table. Detailed mediation analysis output.**
(DOCX)

**S4 Table. Comprehensive E-value sensitivity analysis.**
(DOCX)

**S1 Fig. The Healthy Smoker Paradox – Main finding.**
(TIF)

**S2 Fig. Mediation analysis – Nutritional pathways.**
(TIF)

**S3 Fig. Effect modification – Stronger in vulnerable groups.**
(TIF)

**S4 Fig. Sensitivity analysis – Robustness assessment.**
(TIF)

**S5 Fig. Conceptual framework – Fundamental cause theory.**
(TIF)

**S1 Note. Robustness checks and additional analyses.**
(DOCX)

**S1 Methods. Statistical power considerations and variable coding details.**
(DOCX)

**S1 File. Complete bilingual questionnaire (English/Arabic).**
(DOCX)

**S2 File. Comprehensive laboratory protocol and quality assurance.**
(DOCX)

**S3 File. Complete R statistical analysis script.**
(DOCX)

**S4 File. Comprehensive data dictionary.**
(DOCX)

**S5 File. Comprehensive data collection manual.**
(DOCX)

**S6 File. Instrument calibration protocols.**
(DOCX)

**S7 File. Detailed statistical analysis plan.**
(DOCX)

**S8 File. Data cleaning protocol (.R file).**
(DOCX)

**S1 Data. Raw data anonymized.**
(CSV)

## Author contributions

**Conceptualization:** Radfan saleh Abdullah.

**Data curation:** Radfan saleh Abdullah.

**Formal analysis:** Naif Taleb Ali.

**Funding acquisition:** Radfan saleh Abdullah.

**Investigation:** Radfan saleh Abdullah.

**Methodology:** Naif Taleb Ali.

**Project administration:** Radfan saleh Abdullah.

**Resources:** Mansour Abdelnabi H.Mehdi.

**Software:** Radfan saleh Abdullah, Mansour Abdelnabi H.Mehdi.

**Supervision:** Radfan saleh Abdullah.

**Validation:** Naif Taleb Ali, Mansour Abdelnabi H.Mehdi.

**Visualization:** Naif Taleb Ali, Mansour Abdelnabi H.Mehdi.

**Writing – original draft:** Radfan saleh Abdullah.

**Writing – review & editing:** Naif Taleb Ali, Mansour Abdelnabi H.Mehdi.

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
