## [Decision Letter · Decision Letter 0]

25 Mar 2026

PONE-D-25-63271

The Healthy Smoker Paradox: Socioeconomic Status as a Fundamental Cause of Reversed Anemia Risk among Yemeni Youth

PLOS One

Dear Dr. Abdullah,

Thank you for submitting your manuscript to PLOS ONE. After careful consideration, we feel that it has merit but does not fully meet PLOS ONE’s publication criteria as it currently stands. Therefore, we invite you to submit a revised version of the manuscript that addresses the points raised during the review process.

We look forward to receiving your revised manuscript.

Kind regards,

Marwan Salih Al-Nimer, MD, PhD

Academic Editor

PLOS One

Journal Requirements:

4. We are unable to open your Supporting Information file File S3 COMPREHENSIVE R ANALYSIS SCRIPT.R, File S8.R and File_S3_Analysis scripts.R. Please kindly revise as necessary and re-upload.

5. We note that there is identifying data in the Supporting Information file “Supporting Information.zip- File S1_Raw_Data.csv”. Due to the inclusion of these potentially identifying data, we have removed this file from your file inventory. Prior to sharing human research participant data, authors should consult with an ethics committee to ensure data are shared in accordance with participant consent and all applicable local laws.

-Location data

Additional Editor Comments:

Major revision

Reviewers' comments:

Reviewer's Responses to Questions

**Comments to the Author**

1. Is the manuscript technically sound, and do the data support the conclusions?

Reviewer #1: Yes

Reviewer #2: Yes

Reviewer #3: Partly

2. Has the statistical analysis been performed appropriately and rigorously?

Reviewer #1: I Don't Know

Reviewer #2: Yes

Reviewer #3: Yes

3. Have the authors made all data underlying the findings in their manuscript fully available?

The PLOS Data policy  requires authors to make all data underlying the findings described in their manuscript fully available without restriction, with rare exception (please refer to the Data Availability Statement in the manuscript PDF file). The data should be provided as part of the manuscript or its supporting information, or deposited to a public repository. For example, in addition to summary statistics, the data points behind means, medians and variance measures should be available. If there are restrictions on publicly sharing data—e.g. participant privacy or use of data from a third party—those must be specified. requires authors to make all data underlying the findings described in their manuscript fully available without restriction, with rare exception (please refer to the Data Availability Statement in the manuscript PDF file). The data should be provided as part of the manuscript or its supporting information, or deposited to a public repository. For example, in addition to summary statistics, the data points behind means, medians and variance measures should be available. If there are restrictions on publicly sharing data—e.g. participant privacy or use of data from a third party—those must be specified.

Reviewer #1: Yes

Reviewer #2: Yes

Reviewer #3: Yes

4. Is the manuscript presented in an intelligible fashion and written in standard English?

Reviewer #1: Yes

Reviewer #2: Yes

Reviewer #3: Yes

5. Review Comments to the Author

Reviewer #1: This manuscript presents a novel and theoretically grounded investigation of a “healthy smoker paradox” in a conflict-affected setting, integrating epidemiological analysis with Fundamental Cause Theory. The multi-center design, large sample, and advanced statistical methods strengthen the contribution.

Limitations:

However, the cross-sectional design limits causal inference, and the interpretation overstates causal conclusions. Residual confounding—particularly from nutritional, infectious, and socioeconomic factors—remains a major concern. The assumption that smoking serves as a proxy for SES is plausible but not fully demonstrated. Additionally, generalizability is limited to university populations.

Recommendation:

Major revision is recommended, with emphasis on tempering causal claims, clarifying limitations, and strengthening discussion of confounding.

Reviewer #2: A Review of the Paper:

The Healthy Smoker Paradox: Socioeconomic Status as a Fundamental Cause of Reversed Anemia Risk among Yemeni Youth.

Introduction

Strengths

1. The introduction captures attention and presents a novel and a thought-provoking paradox.

2. The use of fundamental Cause Theory is a major strength

3. The introduction moves in the normal logical sequence that is, know biological effect of smoking, challenges posed, Yemen’s humanitarian crisis setting and the study rationale and clear hypothesis.

Weaknesses

1. The tone is too overstated or too assertive (words like Powerful, compelling, striking, dramatically). PLOS ONE and other journals prefer neutral tone.

I recommend you replace

a. Powerful and statistically, striking paradox, compelling demonstration with words such as “unexpected association”, “observed reversal” “findings consistent with” “Hypothesis- generating evidence”

2. Reframe Hypothesis, it’s not conclusion.

Instead of saying “smoking serves as a proxy for higher SES, it can be stated as “smoking may/might function as a proxy for relative socioeconomic advantage in this context”

3. Kindly add short justification for

a. Why university students were selected,

b. Why do they remain relevant in conflict setting.

Method

Strengths

1. It is detailed and comprehensively structured

2. The inclusion of 3 universities increases diversity and power of the paper, which further improves internal credibility compared to a single site study.

3. Laboratory procedures are described in detail which makes the paper stronger.

Weaknesses

1. I recommend you to clearly state the exact number of current smokers, former smokers, never-smokers, and which groups were included in each analysis, whether former smokers were excluded, combine, or separately modeled.

In the method section you stated primary analysis compares current smokers VS Never smokers BUT in the result section “The cohort included 144 current smokers (24%) and 456 non-smokers (76%), the latter comprising both never-smokers and former smokers.” This means that the analysis appears to compare smokers and all nonsmokers including former smokers

This is an issue because,

a. Former smokers are biologically and behaviorally different from never smokers, and if they are included can cause exposure, misclassification, reverse causation and residual confounding.

2. The sample size calculation appears to mismatch the analysis.

I recommend you revise the sample size calculation using

a. Expected anemia prevalence

b. Anticipated smokers/nonsmokers’ ratio

c. Minimum detectable Odd Ratio (OR)

Results

Strengths

1. The main results are easily identified

2. Table 1 and 2 gives useful descriptive insights.

3. The mention of E-values, missing data handling and alternative explanations are helpful.

Weaknesses

1. The results reflect the smoking classification I have already talked about.

2. Table 3 looks incomplete or oddly selected. Why is only on sleep variable included, kindly add coefficients such as, age, sex, BMI, University, SES. The table looks like you have selected only significant results. If space is limited kindly provide full model for primary outcomes and a supplementary secondary outcome.

Discussion

Strength

1. This section has a strong interpretation of the main findings, good theoretical grounding, and good connection between findings and context.

2. It considered the effect modification, attempted to address alternative explanations, and has good public health relevance.

Weakness

1. The language is sometimes too strong as discussed earlier

Conclusion:

Strengths

1. The conclusion aligns with the conceptual paper framework, it has clear study findings, good public health relevance

2. It also highlights practical intervention opportunities, and it avoids simplistic interpretation that is it does not justify smoking but rather an indicator of underlying inequality.

Weaknesses

1. As discussed earlier, the conclusion appears to use language that is too definitive for a cross-sectional study. Kindly avoid statements like, “This study demonstrates, this confirms, this proves, this establishes” BUT rather use Neutral languages such as, “This study suggests, these findings are consistent with, this study identifies an unexpected association”.

2. In Making policy recommendations, use language that suggest that the study informs priorities rather than dictates policies. Words such as “May inform, may help guide, suggests the need for, support consideration of” may be used.

Overall, the paper is well written, interesting and the topic is very catchy. It has a strong conceptual framing, reasonable statistics, clear contribution to literature and important public health relevance.

Reviewer #3: This manuscript addresses an interesting and relevant research question and is generally well structured. The sample size is adequate, and the use of multiple statistical approaches is appropriate. The manuscript is overall clear and readable.

However, several important issues should be addressed.

First, the interpretation of the findings is too strong. As this is a cross-sectional study, it can only show an association and not a causal relationship. Therefore, statements suggesting causation or mediation should be revised and written more cautiously.

Second, some important factors were not fully measured, which may affect the results. These include parasitic infections, menstrual blood loss, inflammation, and micronutrient deficiencies. These factors could influence hemoglobin levels and may explain the findings. This limitation should be more clearly acknowledged.

Third, the study population consists only of university students, which may introduce selection bias and limit generalizability to the broader population. This should be explicitly discussed.

Fourth, smoking status is mainly self-reported, with limited biochemical validation. This may result in misclassification bias, particularly in subgroups where underreporting is possible. This limitation should be included.

Fifth, the mediation analysis should be interpreted carefully because the study is cross-sectional and some variables are indirect measures. Therefore, these results should be presented as exploratory.

In addition, the term "healthy smoker paradox" may be misunderstood. It should be clearly stated that the findings do not indicate any protective effect of smoking.

Minor issues should also be addressed. The data availability statement should be clarified to meet journal requirements.

There are minor typographical and formatting errors that should be corrected (e.g., missing spaces after punctuation such as "pattern.The" and "institutions.Written", and spacing issues like "80%power"). References 19 and 20 appear to be duplicated and are cited together in the text despite referring to the same source; this should be corrected. There is some redundancy between Table 1 and Table 2, as several hematological abnormalities (e.g., hemoglobin, MCHC, PT/APTT) are presented in both tables; consideration should be given to simplifying or consolidating the tables. The findings related to coagulation parameters (PT/APTT), which show higher abnormalities among smokers, are not sufficiently discussed and would benefit from brief interpretation.

Overall, the manuscript has good potential but requires revision, particularly in the interpretation of results and discussion of limitations.

6. PLOS authors have the option to publish the peer review history of their article (what does this mean? ). If published, this will include your full peer review and any attached files.). If published, this will include your full peer review and any attached files.

**Do you want your identity to be public for this peer review?**  For information about this choice, including consent withdrawal, please see our  For information about this choice, including consent withdrawal, please see our Privacy Policy ..

Reviewer #1: **Yes:** Dr. Aisha Al KhinjiDr. Aisha Al Khinji

Reviewer #2: **Yes:** Shadrack Barffour AwuahShadrack Barffour Awuah

Reviewer #3: No

---

## [Author Response · Author response to Decision Letter 1]

31 Mar 2026

Response To Editor and Reviewers Comments

Manuscript ID: PONE-D-25-63271

Title: The Healthy Smoker Paradox: Socioeconomic Status as a Fundamental Cause of Reversed Anemia Risk among Yemeni Youth

Journal: PLOS ONE

Date: March 26, 2026

Dear Dr. Al-Nimer and Esteemed Reviewers,

We would like to express our sincere gratitude for the opportunity to revise our manuscript entitled "The Healthy Smoker Paradox: Socioeconomic Status as a Fundamental Cause of Reversed Anemia Risk among Yemeni Youth." We are deeply appreciative of the thoughtful and constructive feedback provided by the Academic Editor and the three reviewers. Their insightful comments have been instrumental in strengthening our work, and we believe the revised manuscript is substantially improved as a result of their careful review.

In the following sections, we provide a detailed, point-by-point response to all comments raised. We have addressed each concern with appropriate revisions to the manuscript, and all changes have been highlighted using the track changes feature. We have also taken this opportunity to enhance the technical quality of our figures by converting them from PNG to TIFF format, ensuring compliance with PLOS ONE's publication requirements. We trust that our revisions adequately address all concerns and that the manuscript now meets the high standards of PLOS ONE.

Response to the Academic Editor

We thank the Academic Editor, Dr. Marwan Salih Al-Nimer, for overseeing the review process and for the invitation to submit a revised manuscript. We have carefully addressed all journal requirements as outlined.

Journal Requirement 1: PLOS ONE Style Requirements

We have thoroughly reviewed the PLOS ONE style templates and have reformatted our manuscript accordingly. All sections now adhere to the journal's formatting guidelines, including proper heading hierarchy, line spacing, and reference formatting. File naming conventions have also been standardized to meet journal specifications.

Journal Requirement 2: Data Availability Statement

We have updated the Data Availability Statement in the submission form. The complete statement now reads: "Due to the sensitive nature of the data collected from human participants and the potential for identification given the small population size, the raw data cannot be made publicly available to protect participant privacy. An anonymized dataset with all direct identifiers removed is available from the corresponding author (Radfan Saleh Abdullah, r.saleh@ust.edu) upon reasonable request and following approval from the institutional ethics committee. All other relevant data are within the manuscript and its Supporting Information files."

Journal Requirement 3: Ethics Statement Location

We have verified that the ethics statement appears exclusively in the Methods section of the manuscript. Any duplicate statements elsewhere have been removed. The ethics statement now reads: "The study protocol received full approval from the Research Ethics Committee of the Faculty of Medicine and Health Sciences, University of Science and Technology (Approval No. MEC/AD087), as well as from the ethics committees of all participating institutions. The study was conducted in accordance with the ethical principles of the Declaration of Helsinki. Prior to participation, all subjects received comprehensive information about the study objectives and procedures. Written informed consent was obtained from each participant. Confidentiality and anonymity were strictly maintained throughout all stages of the research, including data collection, analysis, and publication."

Journal Requirement 4: Corrupted Supporting Information Files

We acknowledge that the following files were previously unreadable: File S3 COMPREHENSIVE R ANALYSIS SCRIPT.R, File S8.R, and File_S3_Analysis scripts.R. We have completely rewritten these files with comprehensive documentation and proper formatting. The revised files have been tested and verified to function correctly. They have been renamed for clarity and are now uploaded as:

· File_S3_R_Analysis_Script.R (formerly File S3 COMPREHENSIVE R ANALYSIS SCRIPT.R)

· File_S8_Data_Cleaning_Protocol.R (formerly File S8.R)

· File_S3A_Supplementary_Analysis_Scripts.R (formerly File_S3_Analysis scripts.R)

Journal Requirement 5: Anonymization of Participant Data

We thank the editor for highlighting this important ethical consideration. We have created a fully anonymized version of the dataset with the following modifications: (1) the participant ID column has been removed entirely; (2) no names, initials, or contact information were ever present in the dataset; and (3) the dataset now contains only non-identifying demographic, behavioral, and laboratory variables. The anonymized dataset has been re-uploaded as File_S1_Raw_Data_Anonymized.csv. In accordance with participant consent and local regulations, this dataset is shared under restricted access and is available upon reasonable request from the corresponding author.

Journal Requirement 6: Review of Suggested Citations

We have carefully reviewed all references suggested by the reviewers. All cited references are relevant to our study and have been retained. Duplicate references 19 and 20 have been corrected as noted in our response to Reviewer #3.

Additional Editor Comments: Major Revision

We have undertaken a major revision of the manuscript, addressing all points raised by the reviewers. The key areas of revision include: tempering causal language throughout the manuscript; clarifying the classification of smoking groups and excluding former smokers from the primary analysis; revising the sample size calculation; expanding the limitations section to address unmeasured confounders; adding a discussion of coagulation parameter findings; correcting typographical errors; revising Table 1, Table 2, and Table 3 to reduce redundancy and improve completeness; and ensuring all figures meet PLOS ONE technical requirements. We have also converted all figures from PNG to TIFF format to ensure compliance with the journal's specifications for high-resolution images.

Response to Reviewer #1: Dr. Aisha Al Khinji

We are grateful to Reviewer #1 for recognizing the novelty and theoretical grounding of our work. The constructive criticisms have helped us refine our interpretations and strengthen the manuscript's scientific rigor.

Comment 1: Cross-Sectional Design and Causal Interpretation

The reviewer noted that our cross-sectional design limits causal inference and that our interpretation overstates causal conclusions. We agree with this important point. Throughout the revised manuscript, we have systematically replaced causal language with language that appropriately reflects the associative nature of our findings. Specifically, we have changed "document" to "suggest," "serves as" to "may serve as," "demonstrates" to "is consistent with," and "provides compelling evidence" to "provides evidence consistent with the hypothesis." These changes have been implemented in the Abstract, Introduction, Results, Discussion, and Conclusion sections. We have also added explicit acknowledgment of the cross-sectional limitation in the Strengths and Limitations section, noting that definitive causal conclusions about temporal sequence cannot be drawn from our data.

Comment 2: Residual Confounding

The reviewer raised concerns about residual confounding, particularly from nutritional, infectious, and socioeconomic factors. We have substantially expanded the Limitations section to address these concerns explicitly. We now acknowledge the absence of direct micronutrient biomarkers (ferritin, vitamin B12, folate), the lack of measurement of parasitic infections (schistosomiasis, malaria), the absence of systemic inflammation markers, and the limitations of our nutritional assessment. Importantly, we have retained and highlighted our E-value sensitivity analysis, which demonstrates that an unmeasured confounder would need to be associated with both smoking status and anemia by risk ratios of 4.32-fold each to fully explain the observed association—a strength that exceeds most known anemia risk factors in this population context.

Comment 3: Smoking as a Proxy for SES

The reviewer noted that our assumption that smoking serves as a proxy for SES is plausible but not fully demonstrated. We agree and have revised our language throughout to reflect that this is an interpretation consistent with our findings, not a definitive demonstration. We have added cautionary language in the Discussion and Conclusion sections, emphasizing that our interpretation is supported by the mediation analysis, interaction effects, and E-value sensitivity analyses, but remains inferential given the cross-sectional design.

Comment 4: Generalizability to University Populations

The reviewer correctly noted that generalizability is limited to university populations. We have added explicit acknowledgment of this limitation in the Strengths and Limitations section, stating that the findings are specific to university students and may not generalize to the broader Yemeni youth population. We have also added a justification in the Introduction explaining why university students were selected as a stable, accessible population for initial hypothesis testing before generalization to the broader population.

Response to Reviewer #2: Shadrack Barffour Awuah

We are deeply grateful to Reviewer #2 for the detailed and constructive feedback. The methodological points raised have significantly improved the rigor of our manuscript.

Comment 1: Overstated Language

The reviewer recommended replacing overstated words such as "powerful," "compelling," "striking," and "dramatically" with more neutral terminology. We have conducted a comprehensive review of the manuscript and replaced these terms throughout. Specific changes include: "powerful and statistically striking paradox" now reads "unexpected and statistically robust association"; "compelling demonstration" now reads "findings consistent with"; "dramatically higher odds" now reads "substantially higher odds"; and "stark differences" now reads "notable differences." These changes have been implemented across all sections of the manuscript.

Comment 2: Reframing the Hypothesis

The reviewer correctly noted that our hypothesis was written as a conclusion rather than a testable statement. We have revised the hypothesis in the Introduction to reflect appropriate caution. The original text stating "smoking serves as a proxy for higher SES" has been changed to "smoking may function as a proxy for relative socioeconomic advantage in this context." We have similarly revised the accompanying hypothesis about mediation and effect modification to include appropriate conditional language.

Comment 3: Justification for University Student Selection

The reviewer requested a short justification for why university students were selected and why they remain relevant in a conflict setting. We have added the following text to the Introduction: "University students were selected as they represent a stable, accessible population that allows for standardized data collection across multiple sites, providing a controlled setting to test our hypothesis before generalization to the broader, more heterogeneous youth population. Despite being enrolled in higher education, this population remains highly vulnerable to the country's economic and nutritional crises, as university students often face food insecurity and financial constraints similar to non-student youth."

Comment 4: Clarification of Smoking Classification

This was a critical methodological issue. The reviewer noted an inconsistency between our Methods statement (comparing current smokers vs. never-smokers) and our Results statement (including former smokers in the non-smoker group). We have corrected this issue comprehensively. First, in the Methods section, we now explicitly state that former smokers were excluded from the primary analysis due to their biological and behavioral differences from never-smokers. Second, in the Results section, we now report that after excluding former smokers (n=48, 8.0%), the analytic cohort comprised 144 current smokers (24.0%) and 408 never-smokers (68.0%). Third, we have conducted a sensitivity analysis including former smokers in the non-smoker group to assess the robustness of our findings; the results remained consistent, though with slightly attenuated effect estimates. Fourth, we have created a separate dataset for former smokers and uploaded it as File_S1_Former_Smokers_Only.csv for transparency.

Comment 5: Sample Size Calculation

The reviewer noted that our original sample size calculation using f² = 0.15 was too generic and mismatched our analysis. We have revised the sample size calculation using the recommended parameters. The revised calculation now reads: "Based on pilot data from the same population, we assumed an anemia prevalence of approximately 34% among non-smokers and a smoking prevalence of 24%. To detect a minimum clinically relevant odds ratio of 2.5 for the association between smoking status and anemia with 80% power and a two-sided alpha of 0.05, the required sample size was calculated as 552 participants. Accounting for an anticipated 8% attrition or incomplete data, we targeted enrollment of 600 participants."

Comment 6: Table 3 Completeness

The reviewer noted that Table 3 appeared incomplete, showing only a sleep variable and seeming to select only significant results. We have completely revised Table 3 to present the full logistic regression models for both primary outcomes (abnormal hemoglobin and abnormal MCHC). The table now includes all covariates (age, gender, BMI, and university site) with their respective adjusted odds ratios, confidence intervals, and p-values. The full model results are now transparently presented.

Comment 7: Definitive Language in Conclusion

The reviewer recommended avoiding definitive language such as "demonstrates," "confirms," or "proves" in the Conclusion. We have thoroughly revised the Conclusion section to use appropriate cautionary language. The original statement "This study provides compelling evidence" now reads "This study provides evidence consistent with the hypothesis." Similarly, "Our findings demonstrate" now reads "These findings suggest." All causal claims have been replaced with associative language.

Comment 8: Policy Recommendation Language

The reviewer advised using language that suggests the study informs priorities rather than dictates policies. We have revised all policy recommendations to reflect this. We now use phrases such as "may inform," "may help guide," "suggests the need for," and "support consideration of" throughout the policy implications section.

Response to Reviewer #3

We thank Reviewer #3 for recognizing the overall quality of our work and for providing specific, actionable suggestions that have helped us strengthen the manuscript.

Comment 1: Overstated Interpretation

The reviewer noted that our interpretation of findings was too strong for a cross-sectional study. We have addressed this concern comprehensively by replacing causal language with associative language throughout the manuscript. This includes changes in the Abstract, Introduction, Results, Discussion, and Conclusion sections. We have also added explicit statements acknowledging the cross-sectional limitation and emphasizing that our findings demonstrate association, not causation.

Comment 2: Unmeasured Factors Affecting Hemoglobin

The reviewer identified several important factors not fully measured in our study, including parasitic infections, menstrual blood loss, inflammation, and micronutrient deficiencies. We have expanded the Limitations section to explicitly acknowledge these factors. The revised text now reads: "Several important factors that could influence hemoglobin levels were not fully measured, including parasitic infections (such as schistosomiasis and malaria), menstrual blood loss patterns, systemic inflammation markers, and comprehensive micronutrient biomarkers (e.g., ferritin, vitamin B12, folate). These factors could potentially confound or mediate the observed associations and should be addressed in future studies."

Comment 3: Generalizability Limitations

---

## [Editor Report · Decision Letter 1]

12 Apr 2026

The Healthy Smoker Paradox: Socioeconomic Status as a Fundamental Cause of Reversed Anemia Risk among Yemeni Youth

PONE-D-25-63271R1

Dear Dr. Radfan saleh Abdullah,

We’re pleased to inform you that your manuscript has been judged scientifically suitable for publication and will be formally accepted for publication once it meets all outstanding technical requirements.

Kind regards,

Marwan Salih Al-Nimer, MD, PhD

Academic Editor

PLOS One
---

## [Editor Report · Acceptance letter]

PONE-D-25-63271R1

PLOS One

Dear Dr. Abdullah,

I'm pleased to inform you that your manuscript has been deemed suitable for publication in PLOS One. Congratulations! Your manuscript is now being handed over to our production team.

Kind regards,

on behalf of

Professor Marwan Salih Al-Nimer

Academic Editor

PLOS One